# Antimicrobial Resistance in Sepsis Cases Due to *Escherichia coli* and *Klebsiella pneumoniae*: Pre-Pandemic Insights from a Single Center in Southwestern Romania

**DOI:** 10.3390/healthcare12171713

**Published:** 2024-08-27

**Authors:** Lucian-Ion Giubelan, Alexandru Ionuț Neacșu, Alexandra Daniela Rotaru-Zavaleanu, Eugen Osiac

**Affiliations:** 1Department of Infectious Diseases, University of Medicine and Pharmacy of Craiova, 200349 Craiova, Romania; ligiubelan@yahoo.com; 2Infectious Diseases and Pulmonology ‘Victor Babes’ Hospital, 200515 Craiova, Romania; 3Experimental Research Center for Normal and Pathological Aging, Department of Functional Sciences, University of Medicine and Pharmacy of Craiova, 200349 Craiova, Romania; alexandra.zavaleanu@gmail.com (A.D.R.-Z.); eugen.osiac@umfcv.ro (E.O.); 4Department of Biophysics, University of Medicine and Pharmacy of Craiova, 200349 Craiova, Romania

**Keywords:** sepsis, *Escherichia coli*, *Klebsiella pneumoniae*

## Abstract

Sepsis is an uncontrolled reaction of the body to an infection, and if not effectively treated, it can progress to septic shock, multiple organ failure, and ultimately, death. Objective: To determine the resistance profile of *Escherichia coli* (*E. coli*) and *Klebsiella pneumoniae* (*K. pneumoniae*) strains isolated in sepsis cases diagnosed at the Infectious Diseases Clinic in Craiova, Romania. Methods: The bacteria responsible for sepsis cases were identified using the Vitek 2 Systems version 06.01, which was then employed to assess their antimicrobial susceptibility (Global CLSI and Phenotypic 2017). Results: We have identified 989 patients diagnosed with bacterial sepsis. Among these, 953 cases were caused by Gram-negative rods, with 415 attributed to *E. coli* and 278 to *K. pneumoniae.* High levels of resistance to ampicillin were recorded for *E. coli* strains isolated in sepsis cases (64.6%); adding sulbactam lowers the level of resistance to 41.8%. Resistance to 3rd generation cephalosporins varied between 7.47 and 14.6% and another 3.41 to 11.1% are dose-dependent susceptibility strains. Resistance to carbapenems (i.e., ertapenem, meropenem) is low—2.18–2.42%. More than 95% of the tested *K. pneumoniae* strains were resistant to ampicillin and adding sulbactam as a β-lactamase inhibitor only halves that level. Resistance to 3rd generation cephalosporins varied between 20.7% and 22.5%; resistance levels for *K. pneumoniae* were notably higher than those for *E. coli*. Over 95% of *K. pneumoniae* strains showed resistance to ampicillin, and resistance to 3rd generation cephalosporins varied between 20.7% and 22.5%. Additionally, *K. pneumoniae* exhibited higher resistance to carbapenems (13.7–19.5%) compared to *E. coli* (2.18–2.42%). Conclusions: Antimicrobial resistance levels are generally lower than continental and national data, except for ampicillin and carbapenems (meropenem and ertapenem). *K. pneumoniae* strains are significantly more resistant than *E. coli* strains.

## 1. Introduction

Sepsis, a life-threatening condition caused by the body’s extreme response to infection, remains a significant cause of morbidity and mortality worldwide [1,2,3]. Data from 2017 estimated that approximately 49 million cases of sepsis and 11 million deaths occurred worldwide [4]. According to the Centers for Disease Control and Prevention (CDC), 33% of patients who died in the hospital also developed sepsis [5]. In recent years, the incidence of sepsis appears to be decreasing; however, the mortality rate remains high, with 24% occurring after 30 days and 32% after 90 days [6]. Early recognition of the syndrome is crucial, since it was estimated that 87% of the patients had sepsis or the causative infections before hospitalization [5]. While sepsis can be caused by viruses, fungi, or parasites, bacteria, particularly Gram-positive cocci and Gram-negative rods, represent the most significant etiology of the clinical syndrome [7]. Sepsis treatment is complicated by bacterial resistance to antimicrobials, and the degree of resistance (multidrug-resistant, extensively drug-resistant, and pandrug-resistant) is well-defined [8,9,10]. In 2017, the World Health Organization (WHO) published a list of bacteria for which the development of new antimicrobials is urgently needed [11]; five of these bacteria are Gram-negative rods (the first three of them listed as priority 1—critical), and their resistance to third-generation cephalosporins, carbapenems, and fluoroquinolones is on the focus, while another matter of concern is resistance to polymyxin [12,13]. The rapid rise in antimicrobial resistance has emerged as a formidable challenge in treating bacterial infections, particularly in sepsis cases [14].

Antimicrobial resistance in *E. coli* and *K. pneumoniae* has been a focal point of numerous studies due to its significant impact on public health [15]. Recent research by Ventola emphasizes that antibiotic resistance is a critical challenge in healthcare, necessitating urgent action to develop new treatment strategies and stewardship programs [16]. Additionally, a study by Cassini et al. quantified the burden of antimicrobial resistance in Europe, demonstrating that resistant infections are associated with substantial morbidity and mortality [17].

This study aims to provide a detailed demographic and resistance profile analysis of *E. coli* and *K. pneumoniae* sepsis cases at the ‘Victor Babeș’ Hospital in Craiova. By examining the distribution of these infections across different specimen types and their resistance patterns, we seek to contribute valuable insights that can better inform clinical practice and public health strategies.

This study focuses on the pre-pandemic period, aiming to provide baseline data on antimicrobial resistance in sepsis cases caused by *E. coli* and *K. pneumoniae*. Understanding the resistance patterns in this period can help in comparing and contrasting with post-pandemic data, which may be influenced by changes in healthcare practices, antibiotic usage, and infection control measures brought on by the COVID-19 pandemic.

## 2. Materials and Methods

### 2.1. Study Design and Setting

A descriptive, retrospective study was conducted at the Infectious Diseases Clinic of ‘Victor Babeș’ Hospital in Craiova, Romania, from 1 January 2017 to 31 December 2019. This period was chosen to represent the pre-pandemic era of the Severe Acute Respiratory Syndrome Coronavirus 2 (SARS-CoV-2).

### 2.2. Data Collection

The data retrieved from the patients’ files (demographic, date of sampling, specimen sampled, identified germs, and susceptibility to antimicrobials) were used to create a Microsoft Excel database.

### 2.3. Inclusion Criteria

Patients diagnosed with sepsis (SOFA score ≥ 2) based on the Third International Consensus Definitions for Sepsis and Septic Shock (Sepsis-3) were included [1]. The bacteria involved in sepsis cases were isolated from regular wards.

### 2.4. Microbiological Identification

Bacterial identification and antimicrobial susceptibility testing were performed using the Vitek 2 Systems version 06.01, adhering to Global CLSI and Phenotypic 2017 standards [18,19,20].

### 2.5. Data Analysis

The Multiple Antibiotic Resistance Index (MAR index) was calculated for each strain, with values ranging from 0 to 1 [21]. Statistical analysis was conducted using the two-tailed Chi-squared test with Yates correction and the unpaired *t*-Student test, with significance set at *p* < 0.05 [22]. Microsoft Excel 365, version 2407 (Build 17830.20166, Click-to-Run) was used to perform the calculations.

### 2.6. Ethical Considerations

Informed consent was obtained from all subjects, and the study was approved by the Ethics Committee of the University of Medicine and Pharmacy from Craiova, Romania (nr. 251/06.11.2023). Our hospital adhered to General Data Protection Regulation (GDPR) rules, and all patients’ names were coded to ensure confidentiality.

## 3. Results

Between 2017 and 2019, we identified 989 patients diagnosed with bacterial sepsis (Gram-positive cocci and Gram-negative bacilli). Out of this number, there were 953 cases of sepsis due to Gram-negative rods, with 415 (43%) cases attributed to *E. coli*, 278 (26%) to *K. pneumoniae*, and 260 other species (*Proteus* spp., *Pseudomonas aeruginosa*, *Acinetobacter baumannii*). This study focused on *Escherichia coli* and *Klebsiella pneumoniae* as these pathogens accounted for 72.7% of the total Gram-negative sepsis cases. The remaining 260 cases, though identified as sepsis contributors, involved diverse bacterial species with smaller individual sample sizes. Due to the low occurrence of these species and to maintain statistical significance, they were excluded from the analysis.

Table 1 presents the demographic characteristics of patients with sepsis caused by *E. coli* and *K. pneumoniae*. Noteworthy findings include a significantly higher proportion of females in the *E. coli* group, while males were more predominant in the *K. pneumoniae* group. Additionally, the average age of patients in the *K. pneumoniae* group was higher than in the *E. coli* group. The average age of *E. coli* cases was 53.03 ± 19.31 years (ranging from 18 to 94 years), while in *K. pneumoniae* cases, it was 56.61 ± 15.16 years (ranging from 16 to 88 years), with a significant difference (*p* = 0.0117). The majority of patients in *E. coli* cases were females, accounting for 282 (67.95%), while in *K. pneumoniae* cases, males were predominant, comprising 155 (55.76%) of the cases, with a highly significant difference (*p* < 0.0001). Most patients resided in rural settings in both groups, with 217 (52.29%) in *E. coli* cases and 144 (51.8%) in *K. pneumoniae* cases. Subjects were recruited from the following counties: Dolj (555, 80.09%), Olt (74, 10.66%), Gorj (40, 5.77%), Vȃlcea (13, 1.88%), Mehedinți (8, 1.15%), Argeș (2, 0.29%), and Caraș-Severin (1, 0.14%), covering the southwestern part of Romania. Demographic data of *E. coli* and *K. pneumoniae* are shown in Table 1.

The demographic data in Table 1 present the burden of *E. coli* and *K. pneumoniae* infections in the ‘Victor Babeș’ Hospital. This study observed differences in infection rates between counties. Dolj county had the highest number of sepsis cases (555, 80.09%), while Mehedinți and other smaller counties had significantly fewer cases.

For all patients, the median SOFA score was 2 (ranging from 2 to 3), with no statistically significant differences observed between *E. coli* and *K. pneumoniae* cases. All microbiological samples were collected prior to the introduction of antimicrobials.

Table 2 presents data on the sources of sepsis caused by *E. coli* and *K. pneumoniae*, detailing the number and percentage of cases from various specimen types.

The majority of *E. coli* cases were detected through urine samples (323, 77.83%), whereas *K. pneumoniae* cases predominantly originated from sputum samples (175, 62.95%).

The significant differences in the distribution of cases from sputum and urine between the two pathogens suggest varying primary infection sites and subsequent pathways to sepsis. These findings underscore the importance of targeted diagnostic and treatment strategies based on the underlying source of infection.

Figure 1 presents a heatmap of the antimicrobial susceptibility patterns of *E. coli* isolates. The heatmap indicates the percentage of isolates susceptible, intermediate, and resistant to various antibiotics.

High Resistance: *E. coli* isolates showed high resistance to ampicillin, reflecting common resistance patterns seen globally.Moderate Resistance: there was moderate resistance to cephalosporins such as ceftriaxone and ceftazidime, highlighting the presence of extended-spectrum β-lactamase (ESBL)-producing strains.Low Resistance: the lowest resistance rates were observed for carbapenems like meropenem and imipenem, indicating their continued effectiveness, though vigilance is required due to emerging resistance.

Elevated levels of resistance to ampicillin were noted among *E. coli* strains isolated from sepsis cases (64.6%). The inclusion of sulbactam decreased the resistance level to 41.8%; however, it remains insufficient for empirical treatment in these instances. The antimicrobial susceptibility of *E. coli* strains isolated in sepsis cases is depicted in Figure 1.

Figure 2 displays a heatmap of the antimicrobial susceptibility patterns of *K. pneumoniae* isolates, similar to Figure 1, showcasing their response to different antibiotics.

High Resistance: *K. pneumoniae* exhibited high resistance to β-lactams, including ampicillin and cefazoline, suggesting widespread β-lactamase production.Moderate Resistance: there was moderate resistance to quinolones such as ciprofloxacin, indicating a significant but lesser extent of resistance compared to β-lactams.Low Resistance: susceptibility to carbapenems like meropenem remained relatively high, although the presence of carbapenem-resistant strains is a growing concern.

Over 95% of the tested *K. pneumoniae* strains exhibited resistance to ampicillin, and the addition of sulbactam as a β-lactamase inhibitor only reduced this level by half. Nevertheless, dose-dependent susceptibility increased from 2.67% to 17.3%. Antimicrobial susceptibility of *K. pneumoniae* isolated in sepsis cases is shown in Figure 2.

The average Multiple Antibiotic Resistance (MAR) index of *E. coli* isolated in sepsis cases was 0.21 ± 0.21. The average MAR index of *K. pneumoniae* isolated in sepsis cases was 0.35 ± 0.25 (*p* < 0.0001 as compared with *E. coli* cases); Figure 3 illustrates the relationship between the number of *E. coli* and *K. pneumoniae* strains and specific MAR index values.

There were no statistically significant differences related to patients’ gender or settings. As for age groups, the data are presented in Figure 4.

## 4. Discussion

Sepsis is one of the most critical emergency issues in practicing medicine. Among the top 10 leading global causes of sepsis are lower respiratory tract infections, diarrheal diseases, HIV infection, malaria, and tuberculosis [4]. The highest incidence of sepsis cases is found in Africa, South and Central Americas, and South and Southeast Asia (ranging from 340 cases per 100,000 inhabitants to up to 10 times this number). In Europe (excluding the European part of Russia, Ukraine, and Turkey), the incidence varies between 120 and 270 cases per 100,000 inhabitants [4]. However, estimating sepsis cases and incidence based on the International Statistical Classification of Diseases and Related Health Problems, 10th version (ICD 10) might not be quite accurate. A recent paper evaluating cases from Scania, Sweden suggests that the real incidence is much higher, with about 747 cases per 100,000 habitants [23]. The resistance patterns observed in *E. coli* and *K. pneumoniae* underscore the ongoing challenge of managing infections caused by these pathogens, especially in vulnerable patient populations [24].

Among sepsis cases with identified causes, bacterial etiologies are predominant, with Gram-negative rods (mainly *E. coli*) responsible for 62.2% of cases, while Gram-positive bacteria (primarily *Staphylococcus aureus*) account for 46.8%. Polymicrobial infections are also observed [25]. Our study has found a striking discrepancy between Gram-negative and Gram-positive bacteria causing sepsis, the former being almost 25 times more isolated than the latter. The observed difference could potentially be attributed to our clinic’s patient profile, as cases related to pulmonology (including tuberculosis) are managed in a nearby facility. Tuberculosis (TB) is a major cause of pulmonary infections worldwide. In regions with a high incidence of TB, the disease can substantially impact the overall burden of pulmonary infections. A study highlighted that TB-related complications such as acute respiratory distress syndrome (ARDS), hospital-acquired pneumonia, and sepsis are common in critically ill patients, contributing to high mortality rates in these patients. The high incidence of TB in our region and its management in a dedicated facility could explain the lower numbers of pulmonary infections reported in our study, as our data primarily reflect non-tuberculosis pulmonary infections [26,27]. In our clinic, tuberculosis cases are managed separately in a dedicated facility, which might influence the lower numbers of pulmonary infections reported in this manuscript. The separation ensures that tuberculosis patients receive specialized care and reduces the risk of transmission to other patients. Consequently, the numbers reported in our study primarily reflect non-tuberculosis pulmonary infections, providing a more accurate picture of sepsis cases caused by other bacterial pathogens. This discrepancy aligns with general expectations where Gram-negative bacteria are more frequently isolated in sepsis cases compared to Gram-positive bacteria. For instance, a study reported that Gram-negative bacteria accounted for approximately 62% of sepsis cases, while Gram-positive bacteria accounted for about 47% [28,29]. The current study also confirms that among all cases of Gram-negative bacterial sepsis, *E. coli* is the most frequently isolated pathogen, with *K. pneumoniae* isolates being identified almost 1.5 times less frequently.

In relation to gender, sepsis appears to be more common in males [4]. However, the current study reports approximately 20% more female cases than male cases diagnosed with sepsis. Nevertheless, when considering specific etiologies, *E. coli* and urosepsis are more prevalent in females, while *K. pneumoniae* and respiratory causes of sepsis are more common in males.

High levels of resistance to ampicillin were observed in *E. coli* strains isolated from sepsis cases (64.6%) [30]. The addition of sulbactam reduced the level of resistance to 41.8%, but it remains suboptimal for empirical treatment in such cases. This level of resistance aligns with European data, where the average resistance to ampicillin alone is 57.4%, and it is consistent with the national antimicrobial resistance study CARMIN-ROM 2017, which reported a resistance rate of 68.7% [31,32]. Resistance to 3rd generation cephalosporins varied between 7.47% and 14.6%, with an additional 3.41% to 11.1% being dose-dependent susceptibility strains. This is slightly lower than the European results, which show a resistance rate of 15.1%, and national data, which are around 20% [31,32]. Except for imipenem/cilastatin, to which only a few *E. coli* strains have been tested, resistance to carbapenems (ertapenem, meropenem) is low, ranging from 2.18% to 2.42%. However, European and national data show even lower percentages (0.1% European data vs. 0.39% national data) [31,32]. Antimicrobial policies implemented in our clinic restricted the use of carbapenems and future studies will assess the effect of these rules. The resistance levels for ciprofloxacin and levofloxacin are 22.7% and 23.7%, respectively, slightly lower than continental and national data [31,32]. We have also recorded a low level of resistance to aminoglycosides (1.93 up to 13.7%), the lowest being recorded for amikacin. Polymyxin E is considered the last resort in the fight against Gram-negative rods [33]; we have tested only ten *E. coli* strains, but, worrisome, one of them proved resistant to this antimicrobial. Evolution in time of the resistance levels and the effects of local antimicrobial policies will be assessed by future studies.

More than 95% of the tested *K. pneumoniae* strains were resistant to ampicillin, and adding sulbactam as a β-lactamase inhibitor only halved that level; however, the dose-dependent susceptibility increased from 2.67% to 17.3%. Continental and national data basically skipped this testing due to extremely high levels of resistance. Resistance to 3rd generation cephalosporins varied between 20.7% and 22.5% (depending on the tested antimicrobial), and another 16.8–21.4% of the strains showed dose-dependent susceptibility. This level of resistance is lower than the 31.2–31.7% European data [31]. The resistance levels for meropenem and ertapenem were 13.7% and 19.5%, respectively, which is 2 to 3 times higher than the continental average, raising serious concerns. Of the slightly over 200 tested *K. pneumoniae* strains, the resistance to levofloxacin and ciprofloxacin was 24.5–26.2%, which is lower than EU data and significantly 2.5 times lower than national data [31,32]. Also, aminoglycosides’ level of resistance is comparable with that recorded in Europe (19.8% vs. 22–24%) [31]. Another concerning finding is that we identified two *Klebsiella pneumoniae* strains resistant to polymyxin E, although only 20 isolates were tested for this antimicrobial. It is to be mentioned that our study found out that, for every specific age group, *K. pneumoniae* strains were 1.5 up to 2 times more resistant to antimicrobials as compared with *E. coli* isolates. Understanding the mechanisms underlying antibiotic resistance, such as β-lactamase production, is crucial for developing effective treatment strategies [34].

According to Krumperman PH [21], bacterial strains with an MAR index below 0.2 are acquired from sources with no or little exposure to antimicrobials (i.e., community). MAR indices above 0.2 suggest hospital-related sources. From this perspective, our *E. coli* and *K. pneumoniae* isolates originated almost equally from community and hospital sources. The average MAR index demonstrated that *K. pneumoniae* is significantly more resistant to antimicrobials compared to *E. coli.* An interesting aspect is the variation in the average MAR index across different age groups. Although the number of isolates is limited for each category (*E. coli* and *K. pneumoniae*), it is noteworthy that higher-than-average levels of resistance were observed in very young and very old individuals, while those in middle age exhibited MAR index levels lower than the average. Generally, we would anticipate a consistent increase in the MAR index as an individual’s age increases. This aligns with findings from Naqvi and Drlica, which highlight how fluoroquinolone resistance can complicate treatment strategies and influence resistance trends over time [35]. We will continue to explore this aspect, but the authors’ opinion is that, for our patients, these data should be viewed as a temporal trend and may indicate an increase in the average level of antimicrobial resistance among younger generations compared to middle-aged generations.

The findings from this study reveal distinct patterns in the sources of sepsis caused by *E. coli* and *K. pneumoniae*, as well as their respective resistance profiles. The predominance of *E. coli* in urinary tract infections leading to sepsis is consistent with previous studies [36]. This pathogen’s high incidence in urine samples (77.83%) underscores the need for effective urinary tract infection management to prevent the progression to sepsis.

*K. pneumoniae*, on the other hand, is primarily associated with respiratory tract infections, as indicated by the high proportion of sputum samples (62.95%). This aligns with research by Podschun and Ullmann [37], which highlights the pathogen’s role in pneumonia, particularly in hospital settings. The statistical significance of the differences in sputum and urine sources between the two pathogens (*p* < 0.0001) further supports the need for tailored diagnostic and therapeutic approaches.

The present research is a descriptive and retrospective, single-center analysis, with a relatively small number of cases. Most of our subjects were only from Dolj County. All types of samples from different areas of the clinic have been counted.

### 4.1. Study Limitations

While this study does have some limitations, it still provides important insights into antimicrobial resistance. It is true that being a single-center, retrospective study means the findings might not apply everywhere, as the data came from the Infectious Diseases Clinic of ‘Victor Babeș’ Hospital in Craiova, Romania, and might not reflect broader trends. This study also only covers the pre-pandemic years (2017–2019), so any changes in resistance patterns due to the COVID-19 pandemic are not captured here. Additionally, the sample size, while sufficient for the main analysis, might not be large enough to pick up on smaller differences in resistance patterns or demographic factors. Despite these limitations, this study still offers valuable information that can help guide future research and deepen our understanding of antimicrobial resistance in this specific context.

### 4.2. Implications for Future Research and Practice

The pre-pandemic data serve as a crucial baseline for understanding the impact of the COVID-19 pandemic on antimicrobial resistance [38]. During the pandemic, changes in antibiotic prescribing practices, increased use of broad-spectrum antibiotics, and shifts in hospital infection control protocols may have influenced resistance patterns. By comparing these findings with post-pandemic data, we can assess the impact of the pandemic on antimicrobial resistance and develop more effective strategies for managing sepsis in the future.

This study highlights the importance of continuous monitoring of antimicrobial resistance and the need for targeted interventions to reduce the burden of resistant infections. The demographic data presented underscore the significant burden of *E. coli* and *K. pneumoniae* infections, particularly in rural settings and among older populations. Tailoring infection control and prevention strategies to these demographics can help mitigate the impact of these infections.

The authors plan to use a similar approach for post-pandemic analysis, emphasizing the comparison of resistance patterns before and after the COVID-19 pandemic. This will involve collecting data from the same clinical settings and using consistent methodologies for identifying bacterial strains and assessing antimicrobial susceptibility. The analysis will account for changes in healthcare practices, antibiotic usage, and infection control measures due to the pandemic. By maintaining methodological consistency, the authors aim to provide a clear comparison of how the pandemic has influenced antimicrobial resistance in sepsis cases in future articles.

## 5. Conclusions

Our data reveal that most sepsis cases with an identified cause diagnosed at the ‘Victor Babeș’ Hospital in Craiova, Romania were attributed to Gram-negative rods. *E. coli* and *K. pneumoniae* together accounted for over 70% of these cases. Urinary sources are the predominant cause of sepsis in *E. coli* cases, while respiratory sources are more prevalent in *K. pneumoniae* cases. Antimicrobial resistance levels are generally lower than continental and national data, except for ampicillin and carbapenems (meropenem and ertapenem). *K. pneumoniae* strains exhibit significantly higher resistance compared to *E. coli* stains. There might be an increase in the average level of antimicrobial resistance in the younger generation compared to the middle-aged generation. This study will serve as a foundation for our further investigations into the etiology of sepsis and antimicrobial resistance in the post-pandemic period.

This study provides valuable insights into the demographic characteristics, sources, and antimicrobial resistance profiles of *E. coli* and *K. pneumoniae* sepsis cases at the ‘Victor Babeș’ Hospital in Craiova. The distinct patterns observed underscore the importance of targeted diagnostic and therapeutic strategies. Addressing the high resistance rates requires coordinated efforts across clinical practice and public health domains. Future research should focus on longitudinal studies to monitor resistance trends and evaluate the impact of intervention strategies.

The high resistance rates observed necessitate urgent action to revise treatment guidelines and ensure the judicious use of antibiotics. Clinicians should consider local resistance patterns when selecting empirical therapies for sepsis. A 2023 review emphasized the importance of timely management of sepsis and septic shock. Key recommendations include the prompt administration of empirical antimicrobial treatment, appropriate fluid replacement, and the use of vasoactive agents like norepinephrine to maintain blood pressure. These measures are crucial for reducing mortality rates associated with sepsis [39]. Moreover, enhancing diagnostic capabilities to rapidly identify resistant strains can improve treatment outcomes and reduce the spread of resistant antibiotic infections. Future research should prioritize the development of novel antimicrobial agents and the implementation of robust surveillance systems to monitor resistance trends [40,41].

## Figures and Tables

**Figure 1 healthcare-12-01713-f001:**
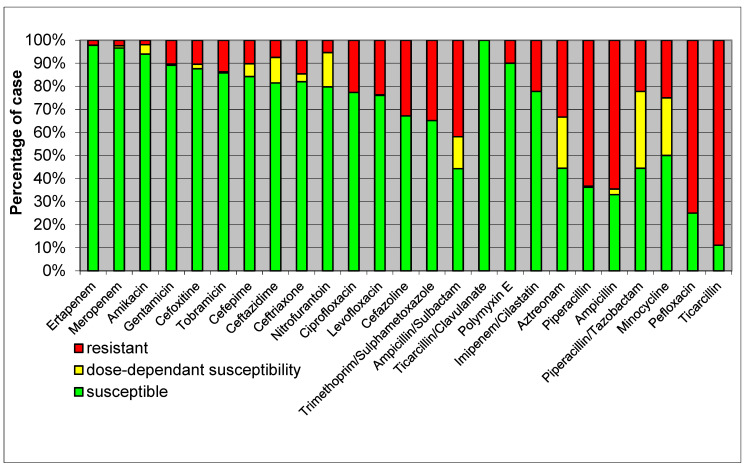
Antimicrobial susceptibility of *E. coli* isolated in sepsis cases. Legend: green = susceptible; yellow = dose-dependent susceptible; red = resistant.

**Figure 2 healthcare-12-01713-f002:**
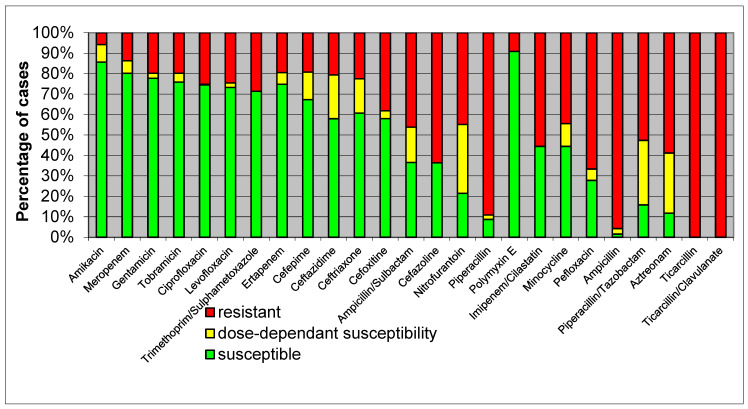
Antimicrobial susceptibility of *K. pneumoniae* isolated in sepsis cases. Legend: green = susceptible; yellow = dose-dependent susceptible; red = resistant.

**Figure 3 healthcare-12-01713-f003:**
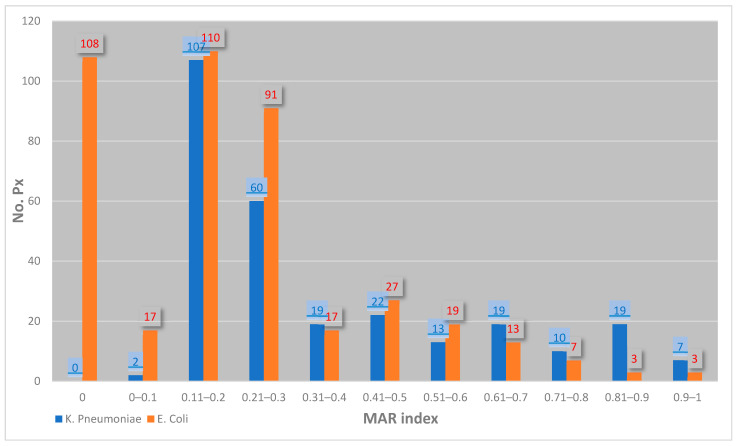
The average Multiple Antibiotic Resistance (MAR) index of *E. coli* and *K. pneumoniae* isolated in sepsis cases. Legend: red = *E. coli*, blue = *K. pneumoniae*. Note: Px = patient.

**Figure 4 healthcare-12-01713-f004:**
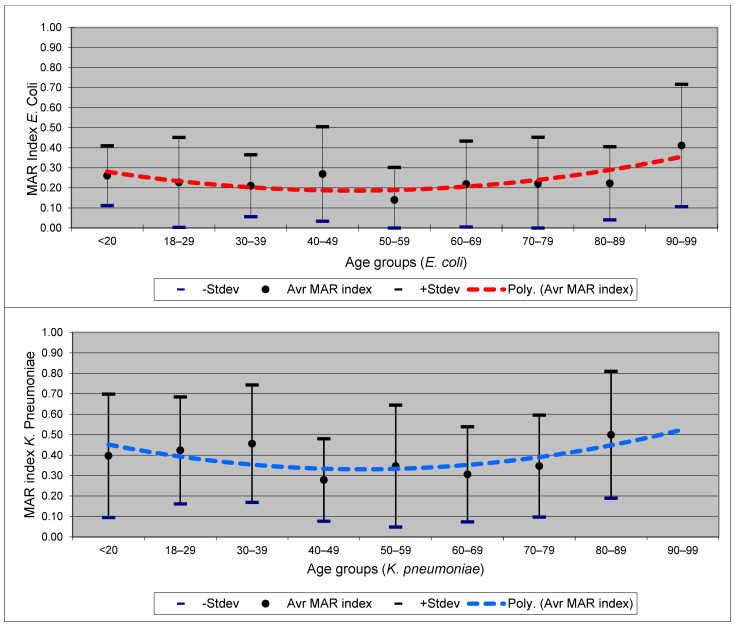
Average MAR index for *E. coli* and *K. pneumoniae* in specific age groups and trendline. Legend: red = *E. coli*, blue = *K. pneumoniae*.

**Table 1 healthcare-12-01713-t001:** Demographic data of *E. coli* and *K. pneumoniae* identified in the ‘Victor Babeș’ Hospital from Craiova.

Characteristic	*E. coli* (n = 415)	*K. pneumoniae* (n = 278)	*p*-Value
Environment			
- Urban	198 (47.71%)	134 (48.20%)	
- Rural	217 (52.29%)	144 (51.80%)	
Age (years)			
- Mean ± SD	53.03 ± 19.31	56.51 ± 15.16	0.0117
- Median (range)	56 (18–94)	59 (16–88)	
Sex			
- Male	133 (32.05%)	155 (55.76%)	
- Female	282 (67.95%)	123 (44.24%)	
- Total	415	278	*p* < 0.001

**Table 2 healthcare-12-01713-t002:** The sources of sepsis due to *E. coli* and *K. pneumoniae* identified in the ‘Victor Babeș’ Hospital from Craiova. N/A: not applicable due to insufficient sample size for statistical analysis.

Specimen	*E. coli* Cases	%	*K. pneumoniae* Cases	%	*p*
Sputum	57	13.73	175	62.95	<0.0001
Bronchial aspirate	0	0.00	4	1.44	N/A
Pus from wound	1	0.24	6	2.16	N/A
Blood	29	6.99	9	3.24	0.0128
Urine	323	77.83	79	28.42	<0.0001
Feces	2	0.48	0	0.00	N/A
Vaginal secretion	2	0.48	2	0.72	N/A
Pericardial fluid	1	0.24	0	0.00	N/A
Cerebrospinal fluid	0	0.00	1	0.36	N/A
Peritoneal fluid	0	0.00	1	0.36	N/A
Bile	0	0.00	1	0.36	N/A
Total	415		278		

## Data Availability

The data presented in this study are available upon request from the corresponding authors.

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
