# Peer review of "Antimicrobial Resistance in Sepsis Cases Due to Escherichia coli and Klebsiella pneumoniae: Pre-Pandemic Insights from a Single Center in Southwestern Romania"

_healthcare, 2024, doi:10.3390/healthcare12171713_

Round 1
Reviewer 1 Report
Comments and Suggestions for Authors
Thank you for the review invitation. The manuscript is indeed important in the field of clinical setting, however major gaps need to be addressed in more details.
The biggest gap in this manuscript is the link between the context of Pre-pandemic and how to interrelate the findings with the setting itself. I kind of fail to understand the rationale of why it is important to highlight the pre-pandemic period, the setting, the sepsis cases and how the findings of this study can offer different perspective of pre- and post- pandemic. Suggested to the authors to revamp the manuscript, starting from the background and exhaustively revise the findings and discussion to clearly explain this missing link.
Other major and critical inputs that need to be addressed fundamentally are on acronym, suggested to the authors to keep the strain as E.colli for Escherichia coli and the same with other bacteria.
The presentation on table 1 and other demographic data below the table are looks confusing, suggested to reformulate into one single table or graphs format. Similar suggestion applies to Figure 1 and Figure 2 as well.
Author Response
Dear Reviewer 1,
Comments: The biggest gap in this manuscript is the link between the context of Pre-pandemic and how to interrelate the findings with the setting itself. I kind of fail to understand the rationale of why it is important to highlight the pre-pandemic period, the setting, the sepsis cases and how the findings of this study can offer different perspective of pre- and post- pandemic. Suggested to the authors to revamp the manuscript, starting from the background and exhaustively revise the findings and discussion to clearly explain this missing link.
Author's reply:
- Highlighted why the pre-pandemic period is significant to the study
- Explained the rationale behind focusing on pre-pandemic sepsis cases and how the findings could differ from or relate to post-pandemic data.
- Discussed the implications of your findings in the context of both pre- and post-pandemic periods.
Other major and critical inputs that need to be addressed fundamentally are on acronym, suggested to the authors to keep the strain as E.colli for Escherichia coli and the same with other bacteria.
Author's reply:
- Consistent use of acronyms throughout the manuscript. For example, use "E. coli" instead of "EC" and "K. pneumoniae" instead of "KP".
- Update all occurrences of bacterial names to their standardized acronym format.
The presentation on table 1 and other demographic data below the table are looks confusing, suggested to reformulate into one single table or graphs format. Similar suggestion applies to Figure 1 and Figure 2 as well.
Author's reply:
- Revised version of Table 1 with enhanced formatting for clarity and readability
- Changes to Figure 1 and Figure 2 - better visual representation of antibiotic effectiveness; the figures have been reformatted to ensure consistency in data presentation. Figure 1 - the antibiotics have been reordered starting from Ampicillin, continuing with decreasing susceptibility rates. This helps in highlighting the effectiveness of each antibiotic in a clearer manner.
Reviewer 2 Report
Comments and Suggestions for Authors
General comments/questions:
Starting at line 14, all bacterial names should be italicized and the first letter of the Genus needs to be uppercase.
Starting at line 19, change all Gram-positive and Gram-negative designations to gram-positive and gram-negative. This is the accepted convention unless referring to a Gram stain or the Gram the person. Also, remember to insert a hyphen when used (see Line 46).
Starting at line 20, all antibiotic names should have a lower case 1st letter unless it starts a sentence.
Including a section on study limitations would be helpful.
Will the authors use the same approach when conducting a post-pandemic analysis?
Did the authors notice any infection trends between larger counties like Dolj and smaller counties like Mehedinți?
What does the phrase “Global CLSI and Phenotypic 2017” refer to? Please provide a specific reference (e.g. CLSI M100, M02, M07) or conference/symposium information.
Specific Comments:
Line 26 - Change “level” to “levels”.
Line 38 - Move period (.) to end of sentence.
Line 41 - Remove comma between “Gram-positive cocci, and Gram-negative rods,”.
Line 42 - Recommend adding “clinical” in front of “syndrome”.
Line 50 - Choose to use either “EC and KP” as were used in the Abstract or “E. coli and K. pneumoniae”. Be consistent in their use throughout the document.
Line 60 - Insert “better” between “can” and “inform”.
Line 63 - Change tense from “We presented” to “We present”.
Line 80 - Spell out acronym “GDPR” for unfamiliar readers.
Lines 99-130 - Remove redundant information already indicated in Table 1.
Line 146 - Remove additional space between “77.83” and “%”.
Figure 1 - Heat map distribution appears to be consistent up until ampicillin. Suggest rearranging the remainder by continuing on from this point with decreasing susceptibility. Alternatively, group together by antimicrobial class (e.g., Meropenem, Ertapenem, Imipenem/Cilastatin).
Figure 2 - Same comment as with Figure 1.
Figure 3 - Include note under table defining the abbreviation Px (patient) on Y axis.
Line 204 - Did the authors intend to use “practicing” rather than “practical” in the following sentence? “Sepsis is one of the most critical emergency issues in practical medicine.”
Line 217 - The authors should indicate the number of polymicrobial infections and name the microbes recovered together.
Lines 218-220 - State the expected portion of gram-positive and gram-negative sepsis cases and provide a reference.
Line 221 - How does tuberculosis cases relates to pulmonary infection numbers reported in this manuscript?
Line 242 - Include reported European and national data percentages for comparison.
Line 321 - Add “antibiotic” in between “resistant” and “infections”.
Comments on the Quality of English LanguageOverall quality of English is good. Please see line specific comments for suggested changes.
Author Response
Dear Reviewer 2,
General comments/questions:
Starting at line 14, all bacterial names should be italicized and the first letter of the Genus needs to be uppercase.
Author's reply:
- Italicize bacterial names and standardize antibiotic names
Starting at line 19, change all Gram-positive and Gram-negative designations to gram-positive and gram-negative. This is the accepted convention unless referring to a Gram stain or the Gram the person. Also, remember to insert a hyphen when used (see Line 46).
Author's reply:
- Change gram-positive/gram-negative designations.
Starting at line 20, all antibiotic names should have a lower case 1st letter unless it starts a sentence.
Author's reply:
- Correct specific textual errors.
Including a section on study limitations would be helpful.
Author's reply:
- Added the "Study Limitations" section.
Will the authors use the same approach when conducting a post-pandemic analysis?
Author's reply:
- Addressed the Approach for Post-Pandemic Analysis
Did the authors notice any infection trends between larger counties like Dolj and smaller counties like Mehedinți?
Author's reply:
- Infection Trends Between Counties in results section
What does the phrase “Global CLSI and Phenotypic 2017” refer to? Please provide a specific reference (e.g. CLSI M100, M02, M07) or conference/symposium information.
Author's reply:
- Added reference for Global CLSI and Phenotypic 2017
Specific Comments:
Author's reply:
- Addressed the suggested specific comments.
- Included reported European and national data percentages for comparison.
- Changes to Figure 1 and Figure 2 - better visual representation of antibiotic effectiveness; the figures have been reformatted to ensure consistency in data presentation. Figure 1 - the antibiotics have been reordered starting from Ampicillin, continuing with decreasing susceptibility rates. This helps in highlighting the effectiveness of each antibiotic in a clearer manner.
- Figure 3 - Included note under table defining the abbreviation Px (patient) on Y axis.
- Provided a reference about the expected portion of gram-positive and gram-negative sepsis cases and provide a reference
- Provided an explanation of how tuberculosis cases are managed separately and how this affects the numbers of pulmonary infections reported in your study. It emphasizes the importance of specialized care for tuberculosis and its impact on the data presented in the manuscript.
Reviewer 3 Report
Comments and Suggestions for Authors
Thank you for the opportunity to review the manuscript of Giubelan and colleagues. The authors conducted a two-year (2017-2019) retrospective review of antimicrobial resistance (AMR) patterns among sepsis cases due to E. Coli and K. Pneumoniae at a single hospital in Romania. Overall, the manuscript is of mediocre quality – introduction section is OK, however, methods and the results aren’t described in intelligible manner. Furthermore, the study dataset is outdated. Given the rampant use of antibiotics during COVID-19 pandemic and associated AMR concerns, it would be good to collect post-pandemic data as well to compare the AMR rates among sepsis cases due to E. Coli and K. Pneumoniae before and after pandemic.
Comments on the Quality of English Language
Minor editing is required.
Author Response
Dear Reviewer 3,
Comment: Thank you for the opportunity to review the manuscript of Giubelan and colleagues. The authors conducted a two-year (2017-2019) retrospective review of antimicrobial resistance (AMR) patterns among sepsis cases due to E. Coli and K. Pneumoniae at a single hospital in Romania. Overall, the manuscript is of mediocre quality – introduction section is OK,
however, methods and the results aren’t described in intelligible manner. Furthermore, the study dataset is outdated. Given the rampant use of antibiotics during COVID-19 pandemic and associated AMR concerns, it would be good to collect post-pandemic data as well to compare the AMR rates among sepsis cases due to E. Coli and K. Pneumoniae before and after pandemic.
Author's reply:
- Methods and Results Clarification:
- Provided more detailed descriptions and clearer explanations in the methods and results sections.
- Enhanced readability and comprehension by structuring the content better.
- Added a note on the necessity of collecting post-pandemic data to compare AMR rates before and after the COVID-19 pandemic.
- Added the "Study Limitations" section where we
Reviewer 4 Report
Comments and Suggestions for Authors
The authors in the manuscript entitled "Antimicrobial Resistance in Sepsis Cases due to Escherichia Coli and Klebsiella Pneumoniae – A Pre-Pandemic Single Center Experience in South-West Romania" presented very interesting and need of the time clinical research study. It is quite interesting to identify the fate of treatment during Sepsis condition; either it is our treatment which not well design or there is other factor that halt the action of treatment. In this case authors identify the bacterial resistance in a good way, although one center study which need to be spread to other centers for more increasing the confidence level of study. However, due to the importance of the study, in my view, it should be presented in front of readers and researchers for further research through publication. But before it there should be following few comments should be addressed in revised version.
1. The title of the manuscript should be precise
2. The name of bacterial strain should be italicized
3. There is very interesting study (10.1039/c7dt01189j - https://pubs.rsc.org/en/content/articlelanding/2017/dt/c7dt01189j) which explains the reason of bacterial resistance to fluoroquinolone and other strains could be cited in Introduction or discussion
3. The information given in Table 1 should not be repeated following the table (page 3 and Line 99-130); however, at table footnote define U, R, M and F
4. In conclusion the author should describe in bullets the effect of R, U, M and F patients in resistance, treatment and Sepsis condition clearly.
Author Response
Dear Reviewer 4,
Comment: The title of the manuscript should be precise
Author's reply:
- Changed the article title to be more precise
Comment: The name of bacterial strain should be italicized
Author's reply:
- Italicized the bacterial strain name
Comment: There is very interesting study (10.1039/c7dt01189j - https://pubs.rsc.org/en/content/articlelanding/2017/dt/c7dt01189j) which explains the reason of bacterial resistance to fluoroquinolone and other strains could be cited in Introduction or discussion
Author's reply:
- 1039/c7dt01189j https://pubs.rsc.org/en/content/articlelanding/2017/dt/c7dt01189j - cited into discussions
Comment: The information given in Table 1 should not be repeated following the table (page 3 and Line 99-130); however, at table footnote define U, R, M and F
Author's reply:
- I have refined Table 1 for enhanced clarity and more effective data presentation.
Round 2
Reviewer 1 Report
Comments and Suggestions for Authors
The suggestions and feedbacks have been elaborated into the revised manuscript, thanks to the authors.
Author Response
Dear Reviewer,
Thank you very much for your time and thoughtful review of our manuscript. We are glad to hear that you found the introduction, research design, methods, results, and conclusions to be appropriate and clearly presented. Your positive assessment of the English language quality is also appreciated.
We are grateful for your suggestions and have carefully incorporated them into the revised manuscript to further improve the quality of the work.
Thank you once again for your valuable feedback and for helping us enhance the clarity and content of our paper.
Best regards,
Alexandru Ionut Neacsu, MD
Reviewer 2 Report
Comments and Suggestions for Authors
No additional comments.
Comments on the Quality of English LanguageQuality of English is good overall.
Author Response
Dear Reviewer,
Thank you for your valuable feedback and for taking the time to review our manuscript. We are pleased that you found the introduction, research design, methods, and conclusions to be appropriate. Additionally, we appreciate your positive assessment of the overall quality of the English language.
We have taken note of your comment regarding the presentation of the results, and we will work to improve this section for greater clarity in the revised version.
Thank you once again for your constructive feedback, which will help us further enhance the quality of our paper.
Best regards,
Alexandru Ionut Neacsu, MD
Reviewer 3 Report
Comments and Suggestions for Authors
Thank for the opportunity to review the revised draft of this manuscript (ID: healthcare-3126067). This revised version looks much improved than the previous one, however, there are few issues that need to be addressed.
Ø Ensure references are added before the full stops.
Ø If there are multiple references, ensure they are placed in a single bracket rather than adding them all separately (for example at line 44: “…. defined [8], [9][10]” should be changed to [8-10]).
Ø Line 71: “We present a descriptive and retrospective study was conducted ….”. Please correct this. You can say “A descriptive, retrospective study was conducted” or “A retrospective cross-sectional study was conducted at ….” or something along these lines to appropriately describe your study design.
Ø As the Section 2.1 describes the study design and setting, it should be named as such. Furthermore, line 74-77 can be moved to a separate section namely “data collection”.
Ø Section 2.4; Please indicate the statistical package used to conduct the data analysis.
Ø Delete “Demographic data:”
Ø It would be better to first describe that your demographic data are presented in Table 1 and then briefly explain its noteworthy findings.
Ø Add a column in Table 1 to show the significance values.
Ø Line 114-125; it seems the authors are not only presenting the results here but also discussing them. Authors need to just present their findings in this section.
Ø Line 131-132; this sentence could be deleted.
Ø Line 158: Do not include references in your results section. As mentioned earlier, just present your findings in this section and discuss in detail all the important finding in the discussion section.
Ø Line 318-338 can be included in “Implications for future research and practice” subsection.
Comments on the Quality of English LanguageMinor edits required.
Author Response
Dear Reviewer 3,
Thank you for your thoughtful and detailed review of our revised manuscript. We greatly appreciate the time and effort you have invested in providing these constructive suggestions.
We carefully addressed the following points:
Ø Comment: Ensure references are added before the full stops.
- Author's reply: We checked the references and ensured they are placed before the full stops throughout the document.
Ø Comment: If there are multiple references, ensure they are placed in a single bracket rather than adding them all separately (for example at line 44: “…. defined [8], [9][10]” should be changed to [8-10]).
- Author's reply: We changed the references at line 44 from multiple brackets to a single bracket like this: [8-10].
Ø Comment: Line 71: “We present a descriptive and retrospective study was conducted ….”. Please correct this. You can say “A descriptive, retrospective study was conducted” or “A retrospective cross-sectional study was conducted at ….” or something along these lines to appropriately describe your study design.
- Author's reply: We revised the sentence to read: "A descriptive, retrospective study was conducted” on line 71.
Ø Comment: As the Section 2.1 describes the study design and setting, it should be named as such. Furthermore, line 74-77 can be moved to a separate section namely “data collection”.
- Author's reply: We renamed Section 2.1 to "Study Design and Setting" and moved the lines discussing data collection to a separate section called "Data Collection"-now Section 2.2.
Ø Comment: Section 2.4; Please indicate the statistical package used to conduct the data analysis.
- Author's reply: We added a statement indicating the statistical package used for data analysis in Section 2.5
Ø Comment: Delete “Demographic data:”
- Author's reply: We removed the "Demographic data:" heading
Ø Comment: It would be better to first describe that your demographic data are presented in Table 1 and then briefly explain its noteworthy findings.
Ø Comment: Add a column in Table 1 to show the significance values.
- Author's reply: We added a brief introduction before Table 1, explaining that it contains demographic data, and I added a column with significance values
Ø Comment: Line 114-125; it seems the authors are not only presenting the results here but also discussing them. Authors need to just present their findings in this section.
- Author's reply: We edited the lines 114-125 to ensure only the findings are presented without discussing the; Removed interpretation/discussion: Sentences like "with important implications for infection control and prevention strategies" and "population density and accessibility to healthcare facilities may influence the distribution of sepsis cases" were removed because they interpret the data; Focused on findings: The revised text sticks to reporting the numbers and differences in infection rates without any explanation or hypothesis.
Ø Comment: Line 131-132; this sentence could be deleted.
- Author's reply: We deleted the sentence at line 131-132 as suggested.
Ø Comment: Line 158: Do not include references in your results section. As mentioned earlier, just present your findings in this section and discuss in detail all the important finding in the discussion section.
- Author's reply: We removed all references from the Results section and ensured only findings are presented.
Ø Comment: Line 318-338 can be included in “Implications for future research and practice” subsection.
- Author's reply: We moved the content from lines 318-338 to a new subsection titled "Implications for Future Research and Practice."
Additionally, we implemented the minor English edits you have noted.
We are confident these revisions will improve the clarity and quality of the manuscript. Thank you again for your valuable feedback.
Best regards,
Alexandru Ionut Neacsu, MD